# Acute high-intensity and moderate-intensity interval exercise do not change corticospinal excitability in low fit, young adults

**Jenin El-Sayes, Claudia V. Turco, Lauren E. Skelly, Mitchell B. Locke, Martin J. Gibala, Aimee J. Nelson***

Department of Kinesiology, McMaster University, Hamilton, Canada

* nelsonaj@mcmaster.ca

## Abstract

Previous research has demonstrated a lack of neuroplasticity induced by acute exercise in low fit individuals, but the influence of exercise intensity is unclear. In the present study, we assessed the effect of acute high-intensity (HI) or moderate-intensity (MOD) interval exercise on neuroplasticity in individuals with low fitness, as determined by a peak oxygen uptake ($VO_{2peak}$) test (n = 19). Transcranial magnetic stimulation (TMS) was used to assess corticospinal excitability via area under the motor evoked potential (MEP) recruitment curve before and following training. Corticospinal excitability was unchanged after HI and MOD, suggesting no effect of acute exercise on neuroplasticity as measured via TMS in sedentary, young individuals. Repeated bouts of exercise, i.e., physical training, may be required to induce short-term changes in corticospinal excitability in previously sedentary individuals.

## Introduction

Transcranial magnetic stimulation (TMS) provides a unique opportunity to non-invasively assess neuroplasticity within the motor system. Single-pulse TMS to the primary motor cortex (M1) can be used to acquire motor-evoked potentials (MEPs), an indicator of corticospinal excitability [1]. One goal of rehabilitation is to alter corticospinal excitability, and this can be measured by changes in the MEP amplitude. Exercise is both cost-effective and can be combined with other rehabilitation protocols to augment the effects of motor re-learning [2]. Numerous studies have used TMS to assess neuroplasticity within the motor system after an acute session of aerobic exercise in healthy individuals (Table 1). These studies have reported either no change [3–7] or an increase [6,8,9] in MEP amplitude following exercise. The discrepancy may relate to either the fitness level of the participants tested, or the intensity of the exercise performed. However, the discrepancy may also relate to the method by which fitness is assessed. For example, the International Physical Activity Questionnaire (IPAQ) is commonly used to assesses physical activity and not fitness *per se*, whereas a peak oxygen uptake ($VO_{2peak}$) test provides an indication of cardiorespiratory fitness. Studies reporting an increase in MEP amplitude after acute exercise were generally performed using highly fit individuals as gauged by a $VO_{2peak}$ test [8,9] or in highly active individuals as gauged by the IPAQ [6]. In

**Funding:** This work was funded by a Natural Sciences and Engineering Research Council of Canada grant (NSERC RGPIN-2015-06309) to AJN and the Canada Research Chairs Program. The funders had no role in the study design, data collection and analysis, decision to publish, or preparation of the manuscript. There was no additional external funding received for this study.

**Competing interests:** The authors have declared that no competing interests exist.

contrast, research performed in low fit, young adults generally found no change in corticospinal excitability following exercise [3–6,10]. Therefore, it appears that fitness level may influence the mechanism by which exercise induces short-term neuroplasticity.

There is also evidence that the magnitude of neuroplastic change may be related to the *intensity* of acute exercise. For example, higher-intensity exercise seemingly induces a greater increase in brain-derived neurotrophic factor (BDNF) and insulin-like growth factor 1 (IGF-1), markers of neuroplasticity [13], compared to lower-intensity exercise [14–19]. Further, improvements in motor skill retention are greater following high- versus low-intensity acute exercise [20]. Opie & Semmler [12] reported an increase in MEPs following both low-intensity continuous and high-intensity interval cycling exercise whereas MacDonald et al. [9] demonstrated increased MEP amplitude following moderate-intensity but not low-intensity continuous cycling. In contrast, other studies have found no change in MEPs following lower versus higher intensity cycling [3,5,11]. These conflicting results demonstrate the discrepancies in the literature relating the effects of exercise intensity on corticospinal excitability. To date, no study has demonstrated changes in MEPs in low fit individuals, providing the incentive to test the effects of exercise *intensity* in this population. A direct comparison of high- versus moderate-intensity exercise is necessary to examine this question and has yet to be performed. Low fit individuals are at a higher risk for experiencing stroke [21]. Therefore, it is important to identify exercise regimes that increase corticospinal excitability this population. Rehabilitation can capitalize on the post-exercise increases in corticospinal excitability, as this can prime the motor system for improvements in motor learning or re-learning [2,22].

In the present study, we tested whether an acute bout of high-intensity (HI) or moderate-intensity (MOD) interval exercise altered corticospinal excitability in low fit individuals. Interval exercise involves short bouts of relatively higher intensity exercise interspersed with brief

**Table 1. Effects of acute cycling on upper limb neurophysiology.**

| Reference | Population | Exercise | MEPs |
|---|---|---|---|
| Singh et al. [4] | n = 12 (5 females, fitness/activity level not reported) | MICT (65–70% age-predicted $HR_{max}$) | ∅* |
| Lulic et al. [6] | n = 14 active (9 females, IPAQ: 7631 ± 6120) n = 14 sedentary (8 females, IPAQ: 1305 ± 773) | MICT (60% age-predicted $HR_{max}$) | ↑ in fit group only# |
| Smith et al. [5] | n = 9 sedentary (4 females, IPAQ: 1784 ± 361) | LICT (40% HRR) M-HICT (80% HRR) | ∅ following both interventions*# |
| Stavrinos & Coxon [10] | n = 24 sedentary (10 females, IPAQ: 2770 ± 1602) | HIIT (90% HRR, 50% HRR) | ∅* |
| McDonnell et al. [3] | n = 25 sedentary (16 females, IPAQ: 1630 ± 906) | LICT (55–65% age-predicted $HR_{max}$) MICT (75% age-predicted $HR_{max}$) | ∅ following both interventions* |
| El-Sayes et al. [8] | n = 34 fit (17 females, $VO_{2peak}$: 46.4 ± 6.6 mL/kg/min) | MICT (65–70% $HR_{max}$) | ↑# |
| MacDonald et al. [9] | n = 15 sedentary-fit (8 females, $VO_{2peak}$: 33.7 ± 7.0 mL/kg/min [range of 22.1–48.2]) | LICT (30% HRR) MICT (40–50% HRR) | ↑ after MICT only* |
| Neva et al. [7] | n = 12 active (6 females, IPAQ: 5112 ± 686) | MICT (65–70% $VO_{2peak}$) | ∅* |
| Andrews et al. [11] | n = 20 sedentary-active (11 females, IPAQ: 4681 ± 2287 | MICT (50% HRR) HIIT (90% HRR, 50% HRR) | ∅ following both interventions# |
| Opie & Semmler [12] | n = 13 (5 females, fitness/activity level not reported) | LICT (50% HRR) HIIT (77% HRR, 25% HRR) | ↑ following both interventions* |

MEPs: motor-evoked potentials; IPAQ: International Physical Activity Questionnaire; $VO_{2peak}$: cardiorespiratory fitness; MICT: moderate-intensity continuous exercise; LICT: low-intensity continuous exercise; HICT: high-intensity continuous exercise; HIIT: high-intensity interval exercise; HRR: heart rate reserve; $HR_{max}$: maximum heart rate; ↓: reductions, ∅: no change; ↑: increases; N/A: not applicable.

*indicates results were obtained immediately post-exercise.

#indicates results were obtained 10-15min post-exercise.

recovery periods [23] and is a potent stimulus for increasing BDNF levels [24]. We chose to use a moderate intensity interval exercise rather than the more commonly used continuous exercise in order to isolate the effect of exercise intensity and remove effects due to the intervallic structure of the exercise. Further, fluctuations in metabolic stress (i.e. an intermittent pattern of exercise) have been shown to influence acute skeletal muscle responses to exercise, independent of exercise intensity [25]. In addition, a recent editorial called for future research to compare physiological responses to moderate-intensity interval exercise and high-intensity interval exercise to better understand the influence of intensity independent of the exercise stimulus pattern (i.e. intermittent) [26]. It was hypothesized that HI would increase corticospinal excitability compared to MOD in low fit young adults, as higher intensity exercise evokes increased BDNF [16–18,27] and IGF-1 [14] levels more so than moderate-intensity exercise.

## Methods

### Participants

Nineteen individuals (22.1 ± 2.6 years; 7 females) participated in three sessions, with a minimum of 48 h between each session. Results from Lulic et al. [6] were used to provide an estimate of the required sample size. The reported effect size for finding a change in MEPs was Cohen's d of 0.5, and assuming a two-tailed alpha of 0.05 and power of 0.8, this yielded a sample size of 22 participants. All individuals reported no history of neurological disease or illness and were right-hand dominant as determined by the modified version of the Edinburgh Handedness Scale [28]. All participants were of low cardiorespiratory fitness, as determined by a $VO_{2peak}$ test, and classification in the "poor" category as defined by the Canadian Society for Exercise Physiology (below 41.6 ml/kg/min for males and 35.0 ml/kg/min for females) [29]. Participants had an average $VO_{2peak}$ of 34.1 ± 4.0 ml/kg/min (coefficient of variation is 11.7%), height of 171.5 ± 9.5 cm, and weight of 69.9 ± 12.6 kg. Participants were also screened for contraindications to TMS [30] and exercise, using a Physical Activity Readiness Questionnaire [31]. Participants were asked to refrain from physical activity on the day of each session, and from consuming alcohol or nicotine for 12 h prior to each session. Written informed consent was obtained prior to participation. This study was approved by the McMaster Research Ethics Board and conformed to the Declaration of Helsinki.

### Experimental design

$VO_{2peak}$ was determined during the first session on an electronically braked cycle ergometer (Lode Excalibur Sport V 2.0, Groningen, the Netherlands) and an on-line gas collection system (Moxus modular oxygen uptake system, AEI Technologies, Pittsburg, PA, USA), as previously described [32]. The $VO_{2peak}$ test began with a warm-up at 50 W for 2 minutes, then the workload was increased by 1 W every 2 seconds until volitional fatigue occurred or until participants could no longer cycle at 60 r.p.m. The $VO_{2peak}$ corresponded to the highest value achieved over a 30 second period. To determine if a valid maximal effort was achieved during the $VO_{2peak}$ test, participants were required to meet two out of four of the following criteria: $HR_{max}$ within 10 bpm of their predicted maximum, respiratory exchange ratio > 1.1, plateau, and/or volitional exhaustion [33]. All participants exerted maximal effort according to these criteria.

Sessions 2 and 3 followed the experimental timeline in Fig 1. Dependent measures were obtained before exercise (T0) and beginning 10 minutes following the end of the exercise intervention (T1). Post-intervention assessments were obtained 10 minutes post-exercise to ensure that heart rate returned to resting levels before data was collected. The order of dependent measure acquisition within each time block was pseudorandomized using the William

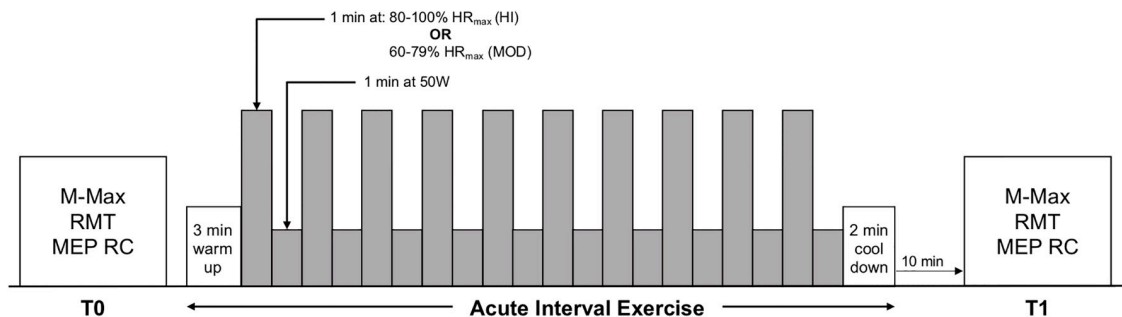

**Fig 1. Experimental timeline.** All dependent measures were acquired before (T0) and beginning ten minutes post-exercise (T1). Dependent measures included maximum M-wave (M-Max), resting motor threshold (RMT), and motor evoked potential (MEP) recruitment curves (RC). The exercise protocols included a 3 minute warm up at 50 W, ten 1 minute bouts interspersed with 1 minute of recovery, and a 2 minute cool down. The intensity of the bouts was 80–100% of maximum heart rate ($HR_{max}$) for HI 60–79% $HR_{max}$ for MOD. Recovery periods involved light cycling at 50 W.

Square Counterbalance. Ten participants (3 females, 7 males) underwent HI in session 2 and MOD in session 3 (described below), while nine participants (4 females, 5 males) underwent MOD followed by HI. Physical activity levels were assessed using the International Physical Activity Questionnaire (IPAQ; [34]) on both experimental sessions to ensure similar physical activity levels were maintained throughout the duration of the study.

## Acute exercise interventions

HI and MOD were performed via lower limb cycling on an electronically braked cycle ergometer (Ergo Race, Kettler, Germany). Both exercise protocols included a 3 minute warm up at 50 W, ten 1 minute bouts interspersed by 1 min recovery periods, and a 2 minute cool down (Fig 1). The intensity bouts was 80–100% of maximum heart rate ($HR_{max}$) for HI and 60–79% $HR_{max}$ for MOD [35] and participants were instructed to cycle between 80–100 r.p.m. The recovery periods for both interventions involved light cycling at 50 W [36], and participants were instructed to cycle at a self-selected pace. Heart rate was monitored using telemetry (Polar A3, New York, USA) to obtain continuous data for the 25 minute exercise period and the 10 minute rest period following the exercise. Ratings of perceived exertion (RPE) were acquired at the end of each interval during the intervention using the 0–10 Borg scale [37]. Throughout the exercise, electromyography (EMG) activity of the right first dorsal interosseous (FDI) muscle ($EMG_{exercise}$) was recorded to ensure that the FDI muscle was inactive.

## Electromyography recording

EMG was recorded from the right FDI using surface electrodes (9 mm diameter Ag-AgCl) placed in a belly tendon montage, with a wet ground electrode placed around the forearm. EMG signals were amplified (x1000), bandpass filtered between 20 Hz and 2.5 kHz (Intronix Technologies Corporation Model 2024F with Signal Conditioning; Intronix Technologies Corporation, Bolton, Canada), and digitized at 5 kHz (Power1401, Cambridge Electronic Design, Cambridge, UK). EMG data were collected using Signal software version 6.02 (Cambridge Electronic Design, Cambridge, UK).

## Maximum M-wave (M-Max)

M-Max was used to normalize MEPs before and after exercise and defined as the maximum response elicited from the right FDI following ulnar nerve stimulation at the wrist. Nerve

stimulation was delivered using a bar electrode (cathode proximal) and a constant current stimulator (Digitimer DS7AH) delivering 200 μs square wave pulses. Stimulation intensity was increased by 1 mA at each trial until the M-wave ceased to increase in 3 consecutive trials. The peak-to-peak amplitude of the M-wave (mV) was defined as M-Max.

### Transcranial magnetic stimulation

Single and paired monophasic TMS pulses were delivered using a custom-built 50 mm diameter figure-of-eight branding coil connected to a Magstim Bistim stimulator (Magstim, Whitland, UK). The TMS coil was positioned 45 degrees in relation to the parasagittal plane to induce a posterior-to-anterior current in the cortex. The motor hotspot for the right FDI was determined within the left motor cortex and defined as the location that elicited large and consistent MEPs. The motor hotspot was digitally registered using Brainsight Neuronavigation (Rogue Research, Canada). RMT was defined as the lowest intensity required to evoke a MEP $\geq$ 50 μV in 5 out of 10 consecutive trials in the relaxed FDI muscle [30]. MEP recruitment curves were obtained from the right FDI muscle at rest by delivering seven TMS pulses at 100–140% RMT in 10% increments in a randomized order (35 pulses total).

### Data analyses

All MEP trials were assessed for background muscle activity. Trials were excluded if the EMG activity immediately before the TMS stimulus artifact exceeded 50 μV [38]. The mean peak-to-peak MEP amplitude at each intensity (100–140% RMT) of the recruitment curve was calculated by averaging the MEPs of the seven trials at each intensity. The Area Under the Recruitment Curve (AURC) was obtained by calculating the trapezoidal integration of the recruitment curve (AURC $= \left( \frac{(MEP_{100\%} + MEP_{110\%})}{2} + \frac{(MEP_{110\%} + MEP_{120\%})}{2} + \frac{(MEP_{120\%} + MEP_{130\%})}{2} + \frac{(MEP_{130\%} + MEP_{140\%})}{2} \right) \times 10$, where $MEP_{100\%}$ is the MEP amplitude at 100% RMT, etc.). AURC was normalized to M-Max (i.e. AURC/M-Max) at T0 and T1 to account for altered electrode conductance that may follow exercise [39].

Group-level analyses included normality testing using the Shapiro-Wilk's test. Outliers were identified using IBM SPSS Software as data points 3 times above or below the interquartile range. No outliers were observed in the data. AURC at T0 was assessed using a Wilcoxon Signed-Rank to determine if AURC at T0 was different between HI and MOD. Since no baseline difference was observed (i.e. T0 in HI was not different than T0 in MOD), AURC was assessed using a repeated-measures ANOVA with factors INTERVENTION (2 levels: HI, MOD) and TIME (2 levels: T0, T1). HI versus MOD effects on IPAQ, RPE, $EMG_{exercise}$, and heart rate were assessed using paired t-tests in cases were data was normally distributed or Wilcoxon Signed-Rank tests in cases where data was not normally distributed. The significance level was set to $p \leq 0.05$ and effect sizes were calculated using Hedge's g.

### Results

All participants were classified as low fit with a mean $VO_{2peak}$ of 34.1 ± 4.0 ml/kg/min (Fig 2). Physical activity levels, assessed via IPAQ, did not differ between the two experimental sessions (HI: 2302.1 ± 2172.3; MOD: 2245.7 ± 2062.5; Wilcoxon: p = 0.65). Exercise details are presented in Table 2. The average HR during the "on" and "off" intervals were significantly different in both the HI (paired t-test, $p < 0.001$, g = 0.85) and MOD exercises (paired t-test, $p < 0.01$, g = 0.41). Further, the $\%HR_{max}$ was significantly different between the "on" and "off" intervals for both the HI (paired t-test, $p < 0.001$, g = 0.86) and MOD exercises (paired t-test, $p < 0.01$, g = 0.38). The HI protocol was more intense than MOD as demonstrated by higher

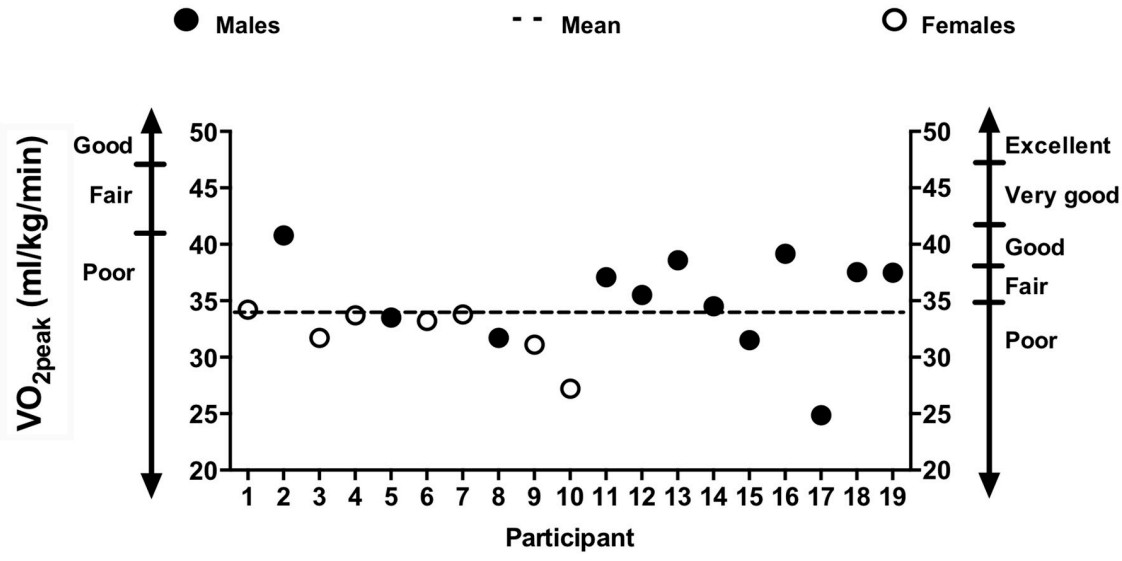

**Fig 2. Fitness distribution of participants.** All participants were classified as sedentary with an average $VO_{2peak}$ of $34.1 \pm 4.0$ ml/kg/min. Our inclusion criteria for 'low fitness' was to achieve a score of "poor" as defined by the Canadian Society for Exercise Physiology (below 41.6 ml/kg/min for males and 35.0 ml/kg/min for females).

heart rate during bouts (HI: $87.1 \pm 6.4\%$ $HR_{max}$; MOD: $70.4 \pm 7.0\%$ $HR_{max}$, paired t-test, $p < 0.001$, g = 2.44) and the greater RPE (HI: $5.5 \pm 1.3$; MOD: $3.5 \pm 1.7$, paired t-test, $p < 0.001$, g = 1.13).

RMT was not different between T0 and T1 for HI (Wilcoxon: p = 0.74, g = 0.02) or MOD (Wilcoxon: p = 0.27, g = 0.04). To assess corticospinal excitability, MEP recruitment curves were obtained and the AURC was calculated. Neither HI or MOD induced a significant change in AURC (Fig 3A; $INTERVENTION_{(1,18)} = 1.07$, $p = 0.31$, $\eta^2 = 0.056$, $TIME_{(1,18)} = 0.50$, $p = 0.49$, $\eta^2 = 0.027$, $INTERVENTION^*TIME_{(1,18)} = 0.01$, $p = 0.92$, $\eta^2 = 0.001$) and there were no differences between HI and MOD at T0 (Wilcoxon: p = 0.42, g = 0.23). There was high between-subject variability in AURC, as shown by the coefficient of variation (HI T0: 67.3%, HI T1: 53.0%, MOD T0: 69.7%, MOD T1: 78.5%). Percent change in AUC (i.e. T0 to T1) for HI and MOD were not different (Fig 3B; Wilcoxon: p = 0.66, g = 0.03). Individual data are depicted in Fig 3C showing variable responses in AURC to HI and MOD.

**Table 2. Exercise details.**

| | HI | | MOD | | Bouts |
|---|---|---|---|---|---|
| | "on" | "off" | "on" | "off" | |
| Heart rate (bpm) | $161.5 \pm 10.8$ | $151.5 \pm 12.3$ | $130.5 \pm 12.0$ | $125.5 \pm 12.0$ | p = 0.001*, g = 2.66 (Wilcoxon) |
| % $HR_{max}$ | $87.1 \pm 6.4$ | $81.6 \pm 6.0$ | $70.4 \pm 7.0$ | $67.7 \pm 7.0$ | $p < 0.001$*, g = 2.44 (paired-t-test) |
| RPE (0–10) | $5.5 \pm 1.3$ | $3.5 \pm 1.9$ | $3.5 \pm 1.7$ | $2.7 \pm 1.9$ | $p < 0.001$*, g = 1.13 (paired t-test) |
| Power (W) | $144.9 \pm 28.8$ | 50 | $78.5 \pm 15.6$ | 50 | $p < 0.001$*, g = 4.26 (Wilcoxon) |
| % of $W_{peak}$ | $68.6 \pm 5.5\%$ | $24.3 \pm 4.3\%$ | $37.3 \pm 4.3\%$ | $24.3 \pm 4.3\%$ | $p < 0.001$*, g = 6.18 (Wilcoxon) |
| $EMG_{exercise}$ | $62.9 \pm 7.6$ | | $62.6 \pm 6.0$ | | p = 0.55, g = 0.05 (Wilcoxon) |

Data are means ± SD. N = 19. g: Hedge's g effect size; HI: Hight-Intensity interval exercise; MOD: Moderate-Intensity interval exercise; bpm: beats per minute; RPE: Ratings of Perceived Exertion; $EMG_{exercise}$: EMG of right FDI during exercise intervention; "on": on intervals; "off": off intervals; $W_{peak}$: peak power

* indicates significance.

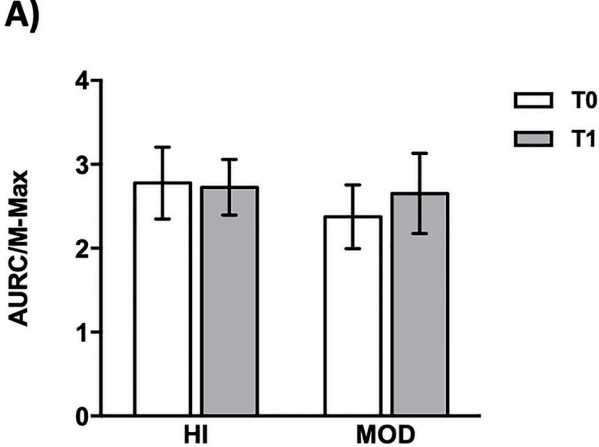

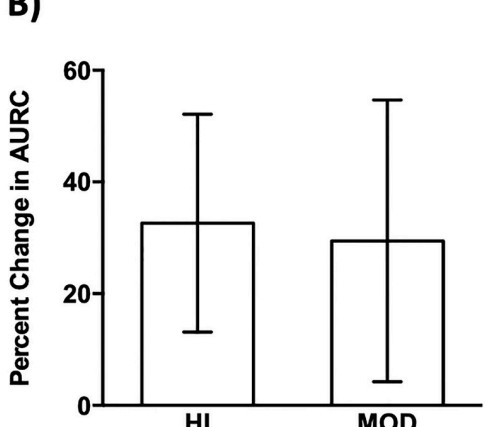

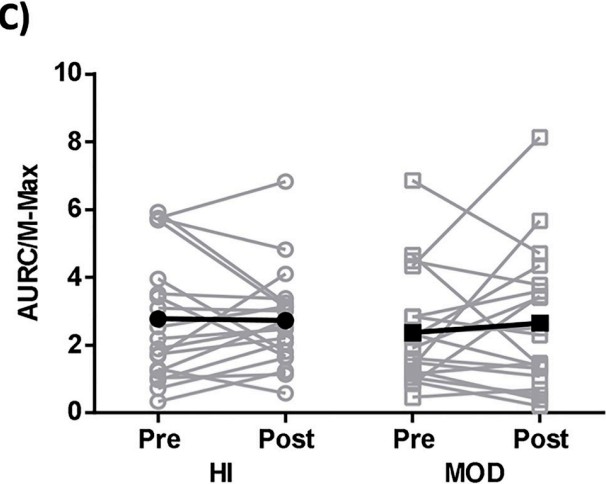

**Fig 3. MEP recruitment curves.** Data are shown as mean ± standard error. **A)** HI and MOD did not induce a significant change in AURC. **B)** Percent change in AURC (i.e. T0 to T1) was not different between HI and MOD. **C)** Individual data showing variable responses in AURC to HI and MOD.

## Discussion

This is the first study to directly compare the effect of acute HI or MOD on exercise-induced neuroplasticity in low fit, young adults. The results suggest that, regardless of intensity, acute exercise does not alter corticospinal excitability.

Our results show that corticospinal excitability was unaltered by either MOD or HI. This is in line with previous work showing no change in corticospinal excitability following moderate- [3–6] and high-intensity exercise [10] in low fit individuals. Increases in corticospinal excitability following exercise have only been observed in high fit individuals after moderate intensity exercise [6,8,9], although some studies have reported to change after moderate intensity exercise [7,11]. Opie and Semmler (2019) also recently showed increased MEP amplitude after both high-intensity interval training and low-intensity continuous exercise, although fitness of the participant sample tested was not reported. There are physiological differences between high and low fit groups that may explain these effects. Compared to low fit groups, high fit participants have greater brain volume [40–42], cerebral blood flow [43–45], and muscle adaptations which may reduce fatigue [46]. Further, high fit individuals show greater levels of IGF-1 [47] and are believed to have greater BDNF uptake into the central nervous system [17], thereby promoting neuroplasticity.

Although we observed no change in corticospinal excitability following HI or MOD in low fit, young adults, it is important to note that these protocols are capable of inducing functional changes in this population. For example, in this population, high-intensity exercise has been shown to improve motor skill consolidation [10], while moderate-intensity exercise reduces reaction time [48], improves memory [49], motor skill acquisition [2,50], and improves motor memory [51]. We note that our findings are limited to the effects of a single session of exercise. It is possible that multiple sessions of MOD or HI may provide a stronger stimulus capable of evoking neuroplasticity in the motor cortex that was not observed following a single bout. However, we note that 6-weeks of high-intensity interval training in low fit individuals did not alter MEPs [52]. Further, it is important to note that these data were obtained from healthy, low fit, young adults. Li et al. [22] recently showed that fast treadmill walking increases MEPs from the lesioned hemisphere in those with chronic stroke. This suggests that high-intensity exercise is a feasible method to increase motor output in stroke rehabilitation. This is in line with research showing that exercise can be used to prime the motor system to facilitate motor learning [53]. While the present study did not show an increase in motor output following high- or moderate-intensity interval exercise, this highlights the importance for research to determine exercise protocols that are capable of increasing motor output in this population.

Although we intended to acquire data from 22 participants, only 19 were available to us. However, a recent study from MacDonald et al. [9] observed an increase in MEPs after moderate intensity cycling in a sample of 15 participants who ranged from sedentary to fit. Therefore, it is unlikely that the lack of effect we observed is due to a limited sample size. One factor that may contribute to the variability in AURC is biological sex. We did not recruit an equal ratio of male to female participants to investigate the effect of biological sex on our data. However, a recent study has reported no effect of biological sex on exercise-induced neuroplasticity. Another factor that may introduce variability is genetic variation. Those with the BDNF val66-met polymorphism show reduced BDNF secretion [54] that is linked to attenuated exercise-induced neuroplasticity responses following high-intensity interval exercise [11] and motor training [55]. We did not determine the distribution of participants presenting the val66met polymorphism, and this is a limitation of the study.

## Conclusions

The present study investigated the effects of exercise intensity on neuroplasticity in young, low fit adults. Corticospinal excitability was assessed before and after HI and MOD. Results revealed that acute exercise did not alter corticospinal excitably, regardless of exercise intensity. Therefore, we conclude that low fit adults do not demonstrate exercise-induced neuroplasticity as measured herein.

## Acknowledgments

We thank Faryal Zahir, Diana Harasym, Prabhav Gogna, and Elizabeth M. Jenkins for their contributions to this work.

## Author Contributions

**Conceptualization:** Jenin El-Sayes, Claudia V. Turco, Lauren E. Skelly, Martin J. Gibala, Aimee J. Nelson.

**Data curation:** Jenin El-Sayes, Claudia V. Turco, Lauren E. Skelly, Mitchell B. Locke.

**Formal analysis:** Jenin El-Sayes, Claudia V. Turco, Aimee J. Nelson.

**Funding acquisition:** Aimee J. Nelson.

**Investigation:** Jenin El-Sayes.

**Methodology:** Jenin El-Sayes, Claudia V. Turco, Aimee J. Nelson.

**Supervision:** Aimee J. Nelson.

**Writing – original draft:** Jenin El-Sayes.

**Writing – review & editing:** Jenin El-Sayes, Claudia V. Turco, Lauren E. Skelly, Mitchell B. Locke, Martin J. Gibala, Aimee J. Nelson.

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
