## [Decision Letter · Decision Letter 0]

24 Oct 2019

PONE-D-19-25228

Acute high-intensity and moderate-intensity interval exercise do not change corticospinal excitability in low fit, young adults

PLOS ONE

Dear Dr. Nelson,

Thank you for submitting your manuscript to PLOS ONE. After careful consideration, we feel that it has merit but does not fully meet PLOS ONE’s publication criteria as it currently stands. Therefore, we invite you to submit a revised version of the manuscript that addresses the points raised during the review process.

Please find below reviews from two expert referees. As you will see, they have some concerns about the novelty of the study and aspects of the study methodology and data reporting. Please ensure all concerns are fully addressed in your re-submission. 

We would appreciate receiving your revised manuscript by Dec 08 2019 11:59PM. To enhance the reproducibility of your results, we recommend that if applicable you deposit your laboratory protocols in protocols.io, where a protocol can be assigned its own identifier (DOI) such that it can be cited independently in the future. For instructions see: http://journals.plos.org/plosone/s/submission-guidelines#loc-laboratory-protocols

We look forward to receiving your revised manuscript.

Kind regards,

Kathryn L. Weston, PhD

Academic Editor

PLOS ONE

Journal Requirements:

'This work was funded in part by a Natural Sciences and Engineering Research Council of Canada grant (NSERC RGPIN-2015-06309) to AJN and the Canada Research Chairs Program.The funders had no role in the study design, data collection and analysis, decision to publish, or preparation of the manuscript.'

Additional Editor Comments (if provided):

Reviewers' comments:

Reviewer's Responses to Questions

**Comments to the Author**

1. Is the manuscript technically sound, and do the data support the conclusions?

Reviewer #1: Yes

Reviewer #2: Partly

2. Has the statistical analysis been performed appropriately and rigorously? 

Reviewer #1: Yes

Reviewer #2: Yes

3. Have the authors made all data underlying the findings in their manuscript fully available?

Reviewer #1: Yes

Reviewer #2: Yes

4. Is the manuscript presented in an intelligible fashion and written in standard English?

Reviewer #1: Yes

Reviewer #2: Yes

5. Review Comments to the Author

Reviewer #1: This study examined the acute effects of moderate and high intensity interval exercise on neuroplasticity in 19 low fit individuals. The paper is well written, and the methods appear to be robust and well executed. I have mainly minor comments and questions for the authors to consider.

Introduction

- In the introduction, I would suggest that the authors need to provide some context of the importance of neuroplasticity and why it is important / necessary to understand the effects of acute exercise on this variable. Because there is limited context provided at present, after reading the manuscript I was left wondering: ‘well, so what?’.

- Line 59-62. The International Physical Activity Questionnaire does not necessarily indicate ‘fitness’, but instead levels physical activity. It is possible to score highly for physical activity but have a relatively poor level of cardiorespiratory fitness, and vice versa. Please consider rewriting this section (and abstract) and modifying table 1 to reflect this.

- Add MEPS acronym to the key in table 1.

- The studies reported in table 1 all appear to have used relatively small sample sizes and one wonders whether the mixed findings are in part related to low statistical power. Is this worth commenting on? I also ask the authors to provide justification of the anticipated statistical power of their own sample size in the methods, given the mixed findings in previous experiments.

- In addition, were MEPS assessed at the same time point post-exercise in these studies? And what was the male/female balance in the study populations?

Methods

- Line 92. Please add all relevant participant characteristics here, e.g. height, mass, VO2peak etc. rather than within figure legends.

- Line 102. Were any criteria employed to determine if a valid maximal effort was achieved during the VO2peak test?

- Were any other controls, other than assessing physical activity, put in place prior to each experimental session? For example, nutritional standardisation, restriction of caffeine etc.? Please add.

- Can the authors provide justification for why 10 minutes post-exercise is most appropriate for assessment of MEPS? This would be useful to add to the paper.

- Similarly, can the authors comment and justify why a comparison between high and moderate intensity interval exercise was chosen as most appropriate. Moderate intensity interval exercise is not typically seen in the literature; it’s more usual to compare HIIT with moderate intensity continuous exercise.

- Given the individual differences in responses shown in figure 2, I’m curious about the reliability of this measurement techniques. Do you have data on this, e.g. coefficients of variation? Please report.

- Can the authors just clarify in the statistics under which circumstances they used paired t-tests or Wilxcoxon signed rank tests.

Results

- Please present IPAQ data in text or in table 2.

- I’m curious about the heart rate data in table 2. There appears to only very small differences in heart rate between on the lower and higher intensity intervals within each bout, much smaller than I would expect especially for HIIT. Is this because you are averaging the heart rate for each minute? If so, it may be better to report a HR during the final 10 seconds of each phase, to better reflect the fluctuations in exercise intensity. For example, based on the HR data, the MOD could be described as continuous rather than interval exercise.

- I find it difficult to pick out meaningful information in figure 3C. I understand it’s purpose, but can I suggest presenting this individual response data in a different way, perhaps as individual lines overlain on a bar chart (with bars not shown), or as a dot plot with each dot representing an individual change score from pre- to post-exercise? Either of these options would make it much easier to pick out the variability in response pre- to post exercise.

Discussion

- Similar to the introduction, please add some context to the discussion r.e. the importance of neuroplasticity, why it matters, and why it is important and noteworthy that you have found no effect of acute exercise in low fit individuals.

- Line 227-229. Please consider rewriting this sentence. It is odd to say that an effect has only been observed in certain circumstances, but then follow it with ‘although not always’.

- Please add a section on the limitations of your experiment including considerations of statistical power and reliability of measurement techniques.

Reviewer #2: This study investigated the effects of two different intensities (moderate and high) of acute exercise on corticospinal excitability. The manuscript addresses a gap in the literature, which concerns the comparison of moderate and high intensity exercise in low fit individuals. The authors could neither find any changes in corticospinal excitability nor intensity dependent changes. It remains unclear why this is the case, hence, precluding any firm interpretation (e.g. interference with design issue.

Although the design fills a gap in the literature, I additionally recommend the authors to expand on the background and introduction by providing more justification for supporting the study design. For example, what is the theoretical relevance of understanding intensity differences in low fit individuals? What is the scientific underpinning leading to the current predictions?

I do have some concerns regarding the current report:

Introduction:

• It would be helpful to add a more detailed section on existing studies of varying intensity load on corticospinal excitability.

• Previous studies, which are also cited here in this manuscript, have examined low-fit subjects to investigate corticospinal excitability after acute exercise. These studies investigated low, moderate and high intensity, albeit in some cases in separate studies. The study by Smith et al also compared two exercise intensities (“low” and “moderate to high”) and could not detect any changes. Although, a direct comparison of moderate and high intensity exercise was not yet investigated, the report would be greatly improved if the derivation of the hypothesis would be described in more detail. The authors need to clarify the new insights and knowledge of their study. This is very important.

Methods:

• The authors report a study design with three sessions. I guess these were performed on separate days. It would be helpful to know the time range in which these sessions were performed (especially with regard to possible carry over effects).

• Alcohol and coffein before exercising can have an influence on the physiological capacity. Did this study control for these influencing parameters?

• In the first paragraph in the methods section the authors report N=19 subjects that participated in this study. On page 6 line 113 the authors wrote: “Eight participants underwent HI in session 2 and MOD in session 3 (described below), while the other half underwent MOD followed by HI.” This suggests that only 16 subjects were included. Moreover, in Table 2 number of subjects was specified as 20. Please clarify how many subjects were included in this study/analysis.

• The authors report a counterbalanced order of intervention (HI-MOD; MOD-HI) within this population. Was the order of intervention also counterbalanced between sexes?

• On Page 5, line 102/103 the authors report that the VO2peak test was performed on a cycle ergometer “Lode Excalibur Sport V2.0”. However, the interventions were performed on a different cycle ergometer “Ergo Race”. Have the authors ensured that the power of the cycle ergometers was validated/the same, e.g. through calibration?

Results:

• The authors report that the IPAQ was acquired on both examination days to ensure that a similar activity level was maintained throughout the study. Please include the IPAQ results. Was the basic activity level maintained?

• The authors report, that they measured the activity of the FDI muscle during exercise to ensure the muscle was inactive. Please report the results.

• Results how a high variability (Figure 3C). Could this be driven by gender differences? It would make sense to exclude gender differences and to include sex as covariate in the statistical analyses.

Discussion:

• The study reported no effects after acute HI or MOD exercise. This has to be discussed in more detail. The authors should try to clarify why there is no effect. Could this also be an experimental design issue? Could the type of exercise intensity play a role (continuous vs interval). Could this be a sample size issue?

• The authors mention a few possible reasons why there might be differences between low and high-fit subjects (page 11). These also need to be discussed in more detail (how do they influence corticospinal excitability).

Minor:

• The authors should review the whole manuscript for punctuation, grammar and spelling.

• Please introduce the abbreviation “RMT” in the text (page 8, line 165).

6. PLOS authors have the option to publish the peer review history of their article (what does this mean?). If published, this will include your full peer review and any attached files.

Reviewer #1: No

Reviewer #2: No

---

## [Author Response · Author response to Decision Letter 0]

8 Nov 2019

See response to reviews uploaded with submission.

---

## [Decision Letter · Decision Letter 1]

23 Dec 2019

Acute high-intensity and moderate-intensity interval exercise do not change corticospinal excitability in low fit, young adults

PONE-D-19-25228R1

Dear Dr. Nelson,

We are pleased to inform you that your manuscript has been judged scientifically suitable for publication and will be formally accepted for publication once it complies with all outstanding technical requirements.

With kind regards,

Kathryn L. Weston, PhD

Academic Editor

PLOS ONE

Additional Editor Comments (optional):

Reviewers' comments:

Reviewer's Responses to Questions

**Comments to the Author**

1. If the authors have adequately addressed your comments raised in a previous round of review and you feel that this manuscript is now acceptable for publication, you may indicate that here to bypass the “Comments to the Author” section, enter your conflict of interest statement in the “Confidential to Editor” section, and submit your "Accept" recommendation.

Reviewer #1: All comments have been addressed

Reviewer #2: All comments have been addressed

2. Is the manuscript technically sound, and do the data support the conclusions?

Reviewer #1: (No Response)

Reviewer #2: Yes

3. Has the statistical analysis been performed appropriately and rigorously? 

Reviewer #1: (No Response)

Reviewer #2: Yes

4. Have the authors made all data underlying the findings in their manuscript fully available?

Reviewer #1: (No Response)

Reviewer #2: Yes

5. Is the manuscript presented in an intelligible fashion and written in standard English?

Reviewer #1: (No Response)

Reviewer #2: Yes

6. Review Comments to the Author

Reviewer #1: (No Response)

Reviewer #2: (No Response)

7. PLOS authors have the option to publish the peer review history of their article (what does this mean?). If published, this will include your full peer review and any attached files.

Reviewer #1: No

Reviewer #2: No

---

## [Editor Report · Acceptance letter]

6 Jan 2020

PONE-D-19-25228R1 

Acute high-intensity and moderate-intensity interval exercise do not change corticospinal excitability in low fit, young adults 

Dear Dr. Nelson:

I am pleased to inform you that your manuscript has been deemed suitable for publication in PLOS ONE. Congratulations! Your manuscript is now with our production department. 

With kind regards,

on behalf of

Dr. Kathryn L. Weston 

Academic Editor

PLOS ONE